# Optical Flow-Guided 6DoF Object Pose Tracking with an Event Camera

## ABSTRACT

Object pose tracking is one of the pivotal technologies in multimedia, attracting ever-growing attention in recent years. Existing methods employing traditional cameras encounter numerous challenges such as motion blur, sensor noise, partial occlusion, and changing lighting conditions. The emerging bio-inspired sensors, particularly event cameras, possess advantages such as high dynamic range and low latency, which hold the potential to address the aforementioned challenges. In this work, we present an optical flow-guided 6DoF object pose tracking method with an event camera. A 2D-3D hybrid feature extraction strategy is firstly utilized to detect corners and edges from events and object models, which characterizes object motion precisely. Then, we search for the optical flow of corners by maximizing the event-associated probability within a spatio-temporal window, and establish the correlation between corners and edges guided by optical flow. Furthermore, by minimizing the distances between corners and edges, the 6DoF object pose is iteratively optimized to achieve continuous pose tracking. Experimental results of both simulated and real events demonstrate that our methods outperform event-based state-of-the-art methods in terms of both accuracy and robustness.

## CCS CONCEPTS

• **Computing methodologies** → **Computational photography**;
• **Hardware** → *Sensor applications and deployments*; • **Theory of computation** → Computational geometry.

## KEYWORDS

Event Camera, Pose Tracking, Optical Flow, Hybrid Feature

## 1 INTRODUCTION

Object pose tracking is a hot research topic in the multimedia and computer vision communities, with significant real-world applications, such as augmented reality [17], robotic grasping [29], and autonomous navigation [28]. The goal of object tracking is to continuously estimate six degrees of freedom (6DoF) that define the rotation and translation of an object relative to the sensor. Currently, there are numerous visual-based solutions that have been developed. Nonetheless, traditional cameras are limited by various factors like low frame rates and limited dynamic range. Hence, the challenges in object tracking persist, such as drastic lighting change,

*ACM MM, 2024, Melbourne, Australia*

© 2024 Copyright held by the owner/author(s). Publication rights licensed to ACM.
ACM ISBN 978-x-xxxx-xxxx-x/YY/MM
https://doi.org/10.1145/nnnnnnn.nnnnnnn

motion blur resulting from rapid object motion, partial occlusion among objects, and interference from cluttered backgrounds.

A novel type of neuromorphic visual sensor, referred to as an "event camera", shows potential in addressing the aforementioned challenges [35]. In contrast to traditional cameras that capture images at a fixed frame rate, event cameras have smart pixels that perceive variations of the logarithmic brightness, operating asynchronously and independently from one another [15]. Whenever a pixel detects a logarithmic intensity change in the scene that surpasses a certain threshold, it triggers an event [10]. Event cameras possess several advantages, including high temporal resolution, high dynamic range, low latency and minimal power consumption. These advantages make event cameras potentially superior to traditional cameras in challenging tasks, such as tracking fast-moving objects [9, 31]. Nevertheless, the unique asynchronous and discretized nature of events makes it difficult to directly apply traditional vision algorithms to address event-based tracking problems. Therefore, it is crucial to develop alternative algorithms to unlock the potential of event cameras for object tracking. The focus should be on addressing two issues: feature extraction and data association.

Feature extraction involves identifying typical features of the moving object from events, such as corners [33], lines [6], and circles [14]. These event-based features can precisely describe the object's appearance, providing critical geometric information for pose determination. Differing from traditional images, events exclusively report brightness changes asynchronously. Therefore, the feature extraction algorithms should fully leverage the distinctive spatio-temporal and photometric characteristics of events. Most events are predominantly triggered by edges of the moving object [26], thereby accurately capturing its contour. The vertices of edges are notably conspicuous, manifesting in events as corners. Furthermore, for 3D objects, the projection of their edges onto the event plane intuitively reflects their poses. This inspires us to comprehensively utilize multiple features, such as corners and edges, for the continuous 6DoF pose tracking of objects.

Data association refers to the establishment of correspondences between event features of the object and its model, namely the 2D-3D feature matching. For most tracking works, it is typically assumed that the initial pose of objects is known in advance [18], thus transforming data association into a matter of matching and updating 2D-2D features (event or features built from events) at consecutive times. One of the primary challenges for data association of event cameras is the lack of any intensity neighborhoods among asynchronous events, potentially reducing the effectiveness of the feature-descriptor approach. The second challenge lies in significant sensor noise and potential event clutter, which may lead to erroneous feature matching. Optical flow estimation is one of the classical algorithms in computer vision, already utilized in object tracking [5, 27], Visual Odometry (VO) [20], and Simultaneous Localization and Mapping (SLAM) [30]. This motivates us to utilize optical flow to establish connections between events and features.

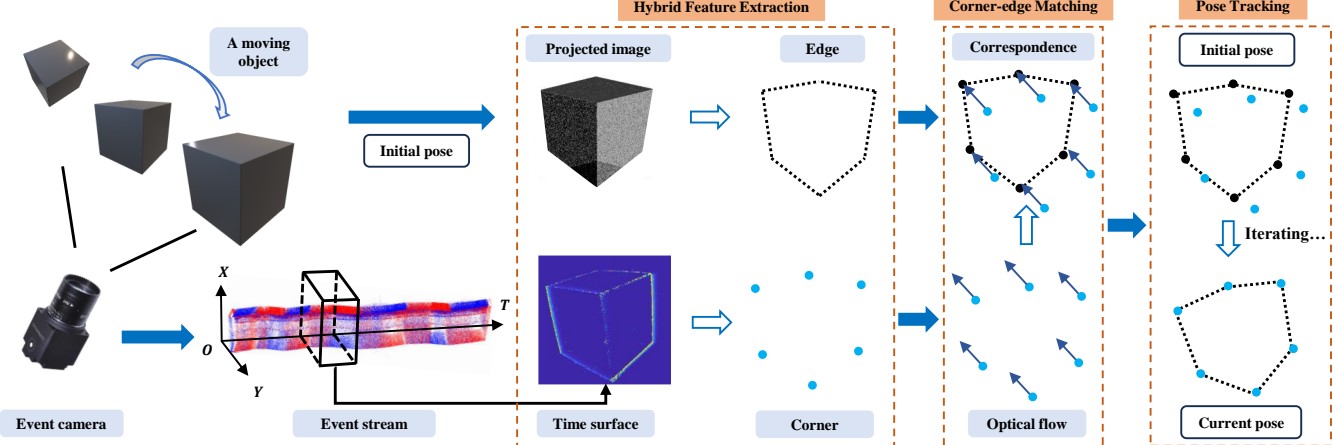

**Figure 1: Flowchart of the proposed method. The core algorithm includes event-based feature extraction, corner-edge matching, and pose tracking. The method takes event streams, initial pose and the object model as input, continuously producing the 6DoF poses of the object as output.**

Upon establishing the feature correspondences, it becomes feasible to formulate an optimization function under the constraint of the perspective projection model, thereby enabling the 6DoF object pose tracking.

In this paper, we propose an optical flow-guided 6DoF object pose tracking method using an event camera, which is summarized in the block diagram of Figure 1. Our method initially employs a hybrid feature extraction strategy, specifically detecting the object corners from the Time Surfaces (TSs) of events, and extracting the object edges from the projected point cloud. Then, we compute the optical flow of corners and search for correspondences with edges along the direction of the optical flow. Finally, we conceptualize object pose tracking as an optical flow-guided iterative optimization problem. The experimental results of simulated and real events indicate that our methods surpass the current state-of-the-art event-based approaches. The main contributions of this work can be summarized as three-fold:

- We proposed a 2D-3D hybrid feature extraction strategy for precisely characterizing object motion, which detects corners and edges from events and point clouds, respectively.
- We model event-based optical flow estimation as a maximization problem of event-associated probability, fully utilizing the spatio-temporal distribution of events.
- We propose an optical flow-guided object pose optimization method, minimizing the distances between corners and edges along the direction of optical flow.

## 2 RELATED WORK

Over the past several decades, there has been extensive research utilizing vision sensors for object pose estimation and tracking. We provide a brief overview of the two primary categories below: monocular-based and event-based object pose tracking.

### 2.1 Monocular-based Object Pose Tracking

Existing monocular pose tracking methods differ in their utilization of edges, features, image regions, direct optimization and deep learning. Feature-based pose tracking methods utilize characteristic features of objects, such as keypoints [23] and lines [19]. Therefore, it is essential to ensure that object surfaces possess ample textures to enable the extraction of a substantial number of features. Region-based methods track objects by maximizing image statistical characteristics between foreground and background regions [25]. Direct methods optimize the photometric error for pose tracking, relying on the fundamental assumption of photometric constancy [7]. Deep learning methods have demonstrated remarkable efficacy in object pose estimation [4]. However, ensuring real-time capability in such data-driven methods remains challenging.

The research on edge-based methods exhibits a high relevance to our current works. The first edge-based object tracking method, RAPID, is proposed by Harris et al. [11], which projects the model edges to images and aligns with image edges. Following the pioneering RAPID, numerous advancements have been put forth to integrate color information or color statistics for precise edge correspondences. Bugaev et al. [1] develop an edge energy function that utilizes both the intensity and orientation of the raw image gradient to determine 2D-3D edge correspondences. While edge-based methods exhibit superior performance for low-textured objects, they frequently encounter challenges when confronted with significant occlusion and densely cluttered backgrounds.

The inherent challenges of monocular-based object pose estimation methods include susceptibility to motion blur and sensitivity to variations in lighting conditions.

### 2.2 Event-based Object Pose Tracking

Early event-based works are very simple, which track moving objects using a known simple shape, such as a blob [8], circle [14] or line [6]. These methods match newly triggered events with the

closest existing blob or feature, and subsequently update the parameters of objects, including their location, size, etc. Nonetheless, the limitation of these methods lies in their effectiveness, which is confined to a narrow range of object shapes. For more complex objects, an event-based iterative closest point algorithm [21] is proposed to update object poses using event-by-event adaptations. Valeiras et al. [22] estimate and track moving objects by establishing connections between events and the 2D projection of objects, given prior knowledge of the wireframe model and initial pose. The recent advancements in machine learning have sparked interest in utilizing data-driven methods for event-based object pose tracking. Jawaid et al. [16] leverage established learning-based approaches to efficiently detect and predict the 2D positions of the satellite. Following this, the Perspective-n-Point (PnP) solver is employed for satellite pose estimation and tracking. Chen et al. [3] propose a novel deep neural network, RM-RNet, which enables the end-to-end estimation of 5-DoF object-level motion, using Time-Surface with Linear Time Decay (TSLTD) frames.

Most existing methods process events by converting them into a frame-like representation to leverage traditional image processing algorithms. However, this process also leads to the loss of temporal and spatial characteristics inherent in events. In this study, we estimate the optical flow by leveraging the spatio-temporal distributed probability of events, and further propose an optical flow-guided pose optimization method.

## 3 METHOD

In this section, the fundamental mathematical concepts and the symbolic representation are elucidated in Section 3.1. After a clear problem definition, an optical flow-guided object pose tracking method is proposed. A hybrid feature extraction method for extracting corners and edges from events and point clouds is presented in Section 3.2. Subsequently, corner-edge matching is employed to establish the correlation between corners and edges guided by optical flow in Section 3.3. Finally, by minimizing the distance between corners and edges, we achieve pose optimization and continuous tracking of objects, as discussed in Section 3.4.

### 3.1 Problem Formulation

Event cameras possess intelligent pixels that independently react to variations in their logarithmic photocurrent $L \doteq \log (I)$. Mathematically, an event can be represented as $e_k = (\mathbf{x}_k, t_k, a_k)$, where $\mathbf{x}_k = (x_k, y_k)^\top$ is pixel coordinates, $t_k$ denotes the trigger time, and $a_k \in \{+1, -1\}$ represents a binary polarity indicating whether the brightness is increasing ("ON") or decreasing ("OFF") [10]. An event is triggered when the brightness alteration of a single pixel surpasses a predetermined threshold, i.e.,

$$\Delta L (\mathbf{x}_k, t_k) \doteq L (\mathbf{x}_k, t_k) - L (\mathbf{x}_k, t_k - \Delta t_k), \quad (1)$$

reaches a temporal contrast threshold $D > 0$, i.e.,

$$\Delta L (\mathbf{x}_k, t_k) = a_k D. \quad (2)$$

Like traditional cameras, event cameras can be characterized by several camera models, with the pinhole camera model being the most widely used among them [14]. To approach this topic mathematically, let us begin with the ideal pinhole camera model.

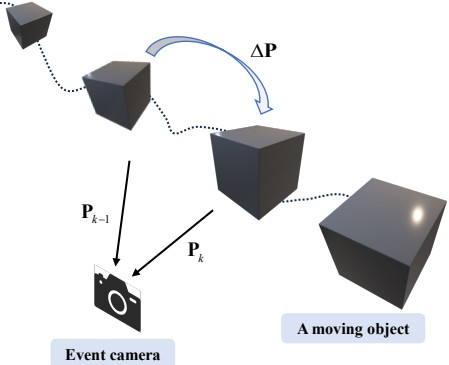

**Figure 2: The geometric interpretation of 6DoF object pose tracking. $P_{k-1}$ and $P_k$ represent the pose of the object at the previous time $t_{k-1}$ and the current time $t_k$ respectively. $\Delta P$ denotes the change in pose from $P_{k-1}$ to $P_k$.**

Suppose that the event $e_k$ is triggered by a 3D point $\mathbf{X}_k$ at time $t_k$. This projection process can be described as

$$s\tilde{\mathbf{x}}_k = \mathbf{KP}_{t_k}\tilde{\mathbf{X}}_k, \quad (3)$$

where $s$ is a scale factor, $\tilde{\mathbf{x}}_k$ and $\tilde{\mathbf{X}}_k$ are the homogeneous coordinates of $\mathbf{x}_k$ and $\mathbf{X}_k$. $\mathbf{K}$ represents the intrinsic parameters of the event camera, which is typically calibrated in advance. $\mathbf{P}$ represents the extrinsic parameters, which describe the relative pose between the object and the event camera.

The event-based pose tracking problem entails continuously solving for the pose $\mathbf{P}$ from event streams, specifically determining the position and orientation of the object relative to the event camera. Inspired by the concept of *recursive pose estimation*, given the object pose $\mathbf{P}_{t_{k-1}}$ at the previous time $t_{k-1}$, the current pose of the object at time $t_k$ can be solved by

$$\mathbf{P}_{t_k} = \mathbf{P}_{t_{k-1}}\Delta \mathbf{P}^{-1}. \quad (4)$$

The geometric interpretation of 6DoF object pose tracking is illustrated in Figure 2. The pose change $\Delta \mathbf{P}$ between adjacent time is relatively small. Therefore, we employ the pose $\mathbf{P}_{t_{k-1}}$ as the initial value and iterate to optimize the current pose $\mathbf{P}_{t_k}$.

### 3.2 Hybrid Feature Extraction

To establish a correlation between 2D images and 3D models, the most common approach involves using 2D-3D feature points [18] or lines [19]. Different from the above methods, we propose a strategy for hybrid feature extraction that fully leverages the distinctive characteristics of events and point clouds. Specifically, it involves extracting corners from events and contours from point clouds, utilizing the most stable feature representations from both modalities.

We transform events into a 2D false-color map, known as the Time Surface (TS), where each pixel records the timestamp of the last event occurring at that location. Utilizing an exponential kernel, TSs prioritize recent events over past occurrences, displaying sensitivity to object edges and the direction of motion [10].

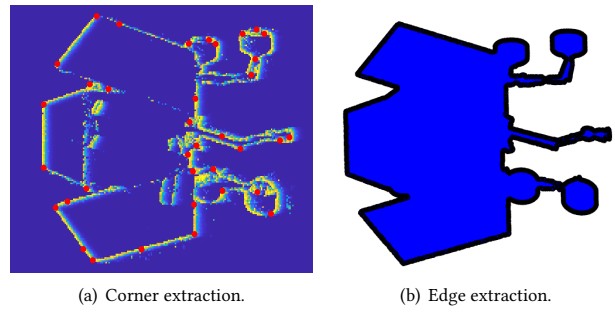

(a) Corner extraction.          (b) Edge extraction.

Figure 3: Feature extraction results. (a) Corners are extracted from TS, indicated in red. (b) Edges are extracted from the projected point cloud (blue), represented in black.

Let $t_{end}$ denote the timestamp of the last triggered event, then the TS at any time $t_k > t_{end}$ is defined as:

$$TS(\mathbf{x}_k, t_k) \doteq e^{-(t_k - t_{end}(\mathbf{x}_k))/\tau}, \tag{5}$$

where $\tau$ is the decay rate parameter, which is set to 30 milliseconds, consistent with [32]. Due to factors such as sensor noise, guided filtering is initially applied to TS for noise reduction while preserving edge features. Then, we detect Harris corners of the object on the filtered TS, as shown in Figure 3(a). We preserve uniformly distributed corners to reflect the current motion state of the object accurately.

Due to occlusion and significant rotations of objects, event-based corner tracking may be unreliable. Hence, we have abandoned the conventional method of establishing correspondences between 2D and 3D feature points, opting instead to directly associate corners with the projected edges of objects. Similar to state-of-the-art monocular pose tracking algorithms, possessing prior knowledge of the object's 3D model and initial pose is an essential prerequisite [18]. Initially, we employ uniform sampling on the 3D model of objects in order to generate a more lightweight point cloud. Following this, utilizing the initial pose, the 3D point cloud of objects can be projected onto the image plane of event cameras. Subsequently, determine the boundaries of the 2D projected point cloud to ascertain the object's edges. Then, retrieve the corresponding 3D edges associated with the detected 2D edges. During each iteration, it is sufficient to project these 3D edges and associate them with corresponding corners. The results of edge extraction are depicted in Figure 3(b), validating the simplicity yet efficacy of our approach.

## 3.3 Corner-edge Matching

Before performing pose tracking, it is essential to address the problem of data association, specifically establishing the correspondences between object corners and edges. The most straightforward approach involves employing brute force matching or nearest neighbors matching to locate the edge points closest to the corners, thereby establishing correspondences. However, this approach may lead to a certain amount of mismatches. We aim to establish effective correspondences with edges through the motion direction of corners, and optical flow provides a promising approach to address this problem. Optical flow estimation is a classical method for

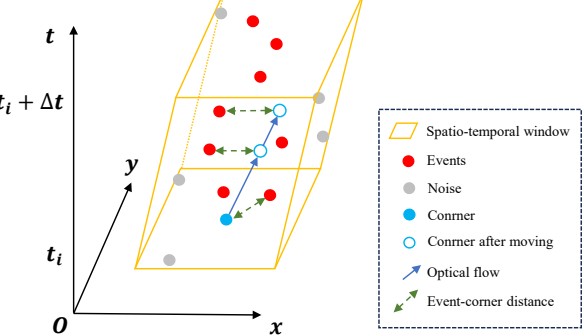

Figure 4: The geometric interpretation of optical flow estimation. The optical flow of corners is computed by establishing correspondences between corners and events within a given spatio-temporal event window.

describing the motion of pixels over time. As time passes and the object moves, corners shift within the event stream. Optical flow can trace the movement of these corners. Along the direction of optical flow, we seek edge points corresponding to these corners.

Let an event corner be denoted as $\mathbf{s}_i = (x_i, y_i)^\top$, and its motion in the event plane can be described using optical flow $\dot{f}(t)$,

$$f(\Delta t) = f(0) + \int_0^{\Delta t} \dot{f}(t)dt = f(0) + \mathbf{u}\Delta t. \tag{6}$$

When the time interval $\Delta t$ is sufficiently small, the average flow $\mathbf{u}$ can be regarded as constant. To seek the optical flow of corners, it is imperative to consider events that are located in the vicinity of corners. The geometric illustration of optical flow estimation is shown in Figure 4. Firstly, we establish an event spatio-temporal window $\{e_k\}_{k=1}^m$ during the time interval $[t, t + \Delta t]$, centered around $\mathbf{s}_i$. Ideally, events within a small window are mostly triggered by the same corner. However, due to factors such as noise, this assumption does not strictly hold true. Inspired by [33, 34], we model the event-based optical flow estimation as a probability distribution problem. Under the guidance of optical flow, the distribution of events triggered by a corner should be more concentrated. We transform the maximization of event probability density into the minimization problem of statistical event-corner distance, modeled as

$$\min_{\mathbf{u}_i} \sum_k^m \sum_i^n \omega_{ki} \left\| \mathbf{x_k} - \left( \mathbf{s}_i + \mathbf{u}_i t_k' \right) \right\|^2 = 0, \tag{7}$$
$$t_k' = t_k - t,$$

where $\omega_{ki}$ represents the correlation probability between the event $e_k$ and the approximate corner $\mathbf{s}_i$ displaced along the optical flow $\mathbf{u}_i$, which should inversely correlate with the distance between them. Taking bisquare weights as an example, it can be expressed as

$$\omega_{ki}(\mathrm{r_{ki}}) = \begin{cases} \left[ 1 - \frac{\mathrm{r}_{ki}^2}{b^2} \right]^2, & |\mathrm{r}_{ki}| < b \\ 0, & |\mathrm{r}_{ki}| \geq b \end{cases}, \tag{8}$$

where $b = 4.685$ is a tuning constant. $\mathrm{r_{ki}}$ is the distance residual between $e_k$ and $\mathbf{s}_i$ after moving along the optical flow $\mathbf{u}_i$. After initializing the weights $\omega_{ki}$, we proceed to solve the linear least

**Algorithm 1** Event-Based Object Pose Tracking

---

**Input:** Events $\{e\}$, Intrinsic matrix $\mathbf{K}$, Initial pose $\mathbf{P}_0$, 3D point cloud of objects $\{O\}$;

**Output:** Object poses $\mathbf{P}_1, \mathbf{P}_2, ..., \mathbf{P}_k, ...$;

1: **for** $t_1, t_2, ..., t_k, ...$ **do**
2:    $\{e_{t_1}\} \leftarrow$ Generate_Event_Window $(\{e\})$;
3:    TS $\leftarrow$ Generate_TS $(\{e_{t_1}\})$, using Eq. (5);
4:    Filtered_TS $\leftarrow$ Guided_Filter (TS);
5:    $\mathbf{s}_i \leftarrow$ Harris_Corner_Detection (Filtered_TS);
6:    **for** $i = 1$ to $n$ **do**
7:      $\omega \leftarrow$ Calculat_weight $(\mathbf{s}_i, \{e_{t_1}\})$, using Eq. (8);
8:      $\mathbf{u}_i \leftarrow$ Calculat_Optical_Flow $(\omega, \mathbf{s}_i, \{e_{t_1}\})$, using Eq. (9);
9:      **if** Meet threshold **then**
10:        Return
11:      **end if**
12:    **end for**
13:    2D_Point_Cloud$\leftarrow$ Project_3D_Point_Cloud $(\{O\}, \mathbf{P}_0, \mathbf{K})$;
14:    $\{\mathbf{c}_i\} \leftarrow$ Extract_Edges (2D_Point_Cloud);
15:    $\{\mathbf{c}_i, \mathbf{s}_i\} \leftarrow$ Corner_Edge_Matching$(\{\mathbf{c}_i\}, \{\mathbf{s}_i, \mathbf{u}_i\})$;
16:    $\mathbf{P}_1 \leftarrow$ Nonlinear_Optimization $(\{\mathbf{c}_i, \mathbf{s}_i\}, \mathbf{P}_0)$, using Eq. (12);
17:    Initial pose $\leftarrow$ Update_pose $(\mathbf{P}_1)$;
18: **end for**

---

squares equation by addressing

$$\mathbf{u}_i \mathbf{A}^\top = \mathbf{B}, \tag{9}$$

where

$$\mathbf{A} = \left[\sqrt{\omega_{11}} t_1', ..., \sqrt{\omega_{ki}} t_k', ...,\right]^\top, \\ \mathbf{B} = \left[\sqrt{\omega_{11}} (\mathbf{x}_1 - \mathbf{s}_2), ..., \sqrt{\omega_{ki}} (\mathbf{x}_k - \mathbf{s}_i), ...,\right]. \tag{10}$$

Then, $u_i$ can be determined by

$$\mathbf{u}_i = \frac{\mathbf{B}\mathbf{A}}{\mathbf{A}^\top \mathbf{A}} = \frac{\sum\limits_{k=1}^{m}\sum\limits_{i=1}^{n} \omega_{ki} (\mathbf{x}_k - \mathbf{s}_i) t_k'}{\sum\limits_{k=1}^{m}\sum\limits_{i=1}^{n} \omega_{ki} t_k'^2}. \tag{11}$$

Through continuous iteration of Eqs. (8) and (9) until the threshold requirement is met or the maximum iteration limit is reached, the final value of $u_i$ is output. Subsequently, corners are propagated along the optical flow direction to locate the corresponding projected edge points $\{\mathbf{c}_i\}$.

### 3.4 Pose Tracking

Given the initial pose, the object model is projected onto the image plane and associated with detected corners by utilizing corner-edge matching. Subsequently, pose optimization is performed based on the corresponding corners and edges, where corners record the object status at the current moment and edge points represent the object pose in the previous moment. By minimizing the distance $d(\cdot)$ between corners and edges, we can optimize the object pose $\mathbf{P}_{t_k}$ at the current time, i.e.,

$$\min_{\mathbf{P}_{t_k}} \sum_{i=1}^{n} d^2 (\mathbf{c}_i, \mathbf{s}_i). \tag{12}$$

For event-based pose tracking, we achieve continuous object tracking by iteratively updating $\mathbf{P}$, which can be represented as

$$\mathbf{P}_{t_k} = \exp\left(\xi^\wedge\right) \in \mathbb{SE}(3),$$
$$\xi^\wedge = \begin{bmatrix} \boldsymbol{\phi}^\wedge & \boldsymbol{\rho} \\ \mathbf{0}^\top & 0 \end{bmatrix} \in \mathbb{R}^{4\times 4}, \xi = \begin{bmatrix} \boldsymbol{\phi} \\ \boldsymbol{\rho} \end{bmatrix} \in \mathbb{R}^6, \boldsymbol{\phi} \in \mathfrak{so}(3), \boldsymbol{\rho} \in \mathbb{R}^3. \tag{13}$$

The pose variation vector $\xi$ comprises both the rotation vector $\boldsymbol{\phi}$ and translation vectors $\boldsymbol{\rho}$. We aim to search for the optimal set of parameters that minimize the objective function (12). This is a challenging problem because the objective function may be highly nonlinear and non-convex, with multiple local minima. To address this issue, one can iteratively update the pose by moving in the direction of the steepest descent. The cost function can be approximated by its first-order Taylor expansion

$$d(\xi + \Delta\xi) \approx d(\xi) + J(\xi)^\top \Delta\xi. \tag{14}$$

The Jacobian matrix of the current pose $\xi$ is

$$\mathbf{J}(\xi) = \frac{\partial d(\xi)}{\partial \xi}. \tag{15}$$

We utilize the Levenberg-Marquardt (LM) algorithm, which restricts the step size of variable updates within a specified trust region, ensuring that the Taylor expansion provides a reliable approximation. We have

$$(\mathbf{H}(\xi) + \lambda\mathbf{I})\Delta\xi = \mathbf{g}(\xi),$$
$$\mathbf{H}(\xi) = \mathbf{J}(\xi)\mathbf{J}(\xi)^\top, \mathbf{g}(\xi) = -\mathbf{J}(\xi)\mathbf{d}(\xi). \tag{16}$$

The increment $\Delta\xi$ is calculated by

$$\Delta\xi = -(\mathbf{H}(\xi) + \lambda\mathbf{I})^{-1}\mathbf{g}(\xi), \tag{17}$$

where $\lambda$ represents the regularization parameter and $\mathbf{I}$ is the identity matrix. We implement this procedure by composing the matrix exponential of the corresponding twist $\Delta\xi^\wedge$ with the previous pose,

$$\mathbf{P}_{t_k} \leftarrow \exp\left(\Delta\xi^\wedge\right)\mathbf{P}_{t_k}. \tag{18}$$

When the threshold is satisfied, we can obtain the optimal object pose $\mathbf{P}_{t_k}$ at the current moment. Upon arrival of new events, leveraging $\mathbf{P}_{t_k}$ as the initial pose, we persistently refine it to attain the new pose $\mathbf{P}_{t_{k+1}}$ at next moment for continuous pose tracking. The overall flow of the proposed method is illustrated in Algorithm 1.

## 4 EXPERIMENTAL EVALUATION

In this section, we assess the effectiveness of our methods using simulated and real events. The experimental details of event collection are provided in Section 4.1. Subsequently, we proceed to execute simulated and real event experiments for method validation as outlined in Sections 4.2 and 4.3, respectively. For further details, please refer to the supplementary materials.

### 4.1 Experimental Preparation

Initially, experiments with simulated events are prepared. We select several representative objects with straight edges (Figure 5(a)), as well as those with curved edges from [12] (Figure 5(b)). Blender is utilized to render RGB videos of object motion, subsequently transforming them into event streams using V2E [13]. To simulate various complex scenarios that objects may encounter during motion, numerous challenging conditions are introduced, such as messy backgrounds, extensive occlusions, and fast movements.

**Table 1: Pose tracking error of objects with straight edges in simulated event experiments ($\delta_{\mathbf{R}}$ :°, $\delta_{\mathbf{T}}$ : $cm$).**

| Sequence | 01 | | | | 02 | | | | 03 | | | |
| --- | --- | --- | --- | --- | --- | --- | --- | --- | --- | --- | --- | --- |
| | Slow | | Fast | | Normal | | Fast | | Slow | | Fast | |
| Method | $\delta_{\mathbf{R}}$ | $\delta_{\mathbf{T}}$ | $\delta_{\mathbf{R}}$ | $\delta_{\mathbf{T}}$ | $\delta_{\mathbf{R}}$ | $\delta_{\mathbf{T}}$ | $\delta_{\mathbf{R}}$ | $\delta_{\mathbf{T}}$ | $\delta_{\mathbf{R}}$ | $\delta_{\mathbf{T}}$ | $\delta_{\mathbf{R}}$ | $\delta_{\mathbf{T}}$ |
| Line-Based | 3.01 | 7.29 | 3.10 | 8.35 | 2.67 | 4.24 | 6.47 | 7.48 | 4.46 | 18.23 | 5.25 | 22.22 |
| LS-Based | **2.19** | 7.10 | 2.86 | 8.10 | 2.41 | **3.67** | 5.69 | 6.24 | 3.96 | 13.25 | 4.54 | 20.44 |
| NNS-Based | 3.11 | 7.82 | 6.95 | 8.54 | 2.95 | 5.18 | 6.94 | 7.38 | 5.95 | 17.15 | 6.12 | 23.52 |
| Ours | 2.24 | **3.42** | **1.94** | **5.56** | **1.53** | 4.47 | **1.75** | **4.43** | **2.12** | **5.44** | **2.65** | **8.56** |
| Sequence | 04 | | | | 05 | | | | 06 | | | |
| | Normal | | Fast | | Normal | | Fast | | Normal | | Fast | |
| Method | $\delta_{\mathbf{R}}$ | $\delta_{\mathbf{T}}$ | $\delta_{\mathbf{R}}$ | $\delta_{\mathbf{T}}$ | $\delta_{\mathbf{R}}$ | $\delta_{\mathbf{T}}$ | $\delta_{\mathbf{R}}$ | $\delta_{\mathbf{T}}$ | $\delta_{\mathbf{R}}$ | $\delta_{\mathbf{T}}$ | $\delta_{\mathbf{R}}$ | $\delta_{\mathbf{T}}$ |
| Line-Based | 1.39 | 2.51 | 2.95 | 7.19 | 0.68 | 7.44 | **1.48** | 8.33 | **1.29** | 4.26 | **1.17** | 5.74 |
| LS-Based | 1.36 | 2.94 | 2.49 | 6.78 | **0.52** | 5.54 | 1.68 | 6.05 | 1.58 | 3.99 | 2.00 | 4.58 |
| NNS-Based | 2.45 | 3.06 | 2.75 | 7.12 | 1.76 | 5.94 | 2.87 | 7.65 | 2.22 | 4.82 | 2.45 | 5.65 |
| Ours | **1.35** | **2.56** | **1.67** | **4.86** | 0.88 | **3.62** | 1.80 | **4.30** | 2.21 | **3.76** | 2.24 | **4.14** |

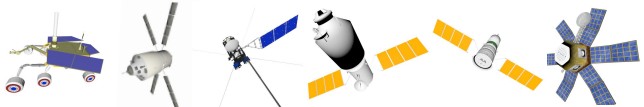

(a) Virtual object with straight edges of simulated event experiments.

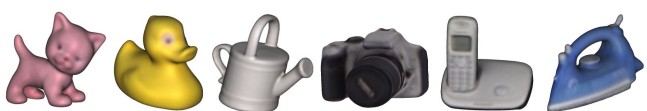

(b) Virtual object with curved edges of simulated event experiments.

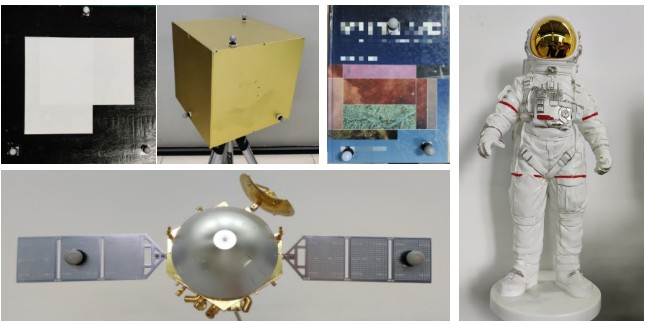

(c) Real object with straight and curved edges of real event experiments.

**Figure 5: Test objects in the experiment. (a) The edges of objects are composed of a considerable quantity of straight lines. From left to right, objects are labeled as 01-06 in sequence. (b) The edges of objects are curved. From left to right, objects are labeled as 07-12 in sequence. (c) The four objects on the left side exhibit straight edges, namely the planar pattern, cube, book, and satellite model, whereas the astronaut model on the right side showcases curved edges.**

To validate the feasibility of the proposed method, experiments with real events are subsequently conducted, including objects with straight and curved edges, as shown in Figure 5(c). Firstly, we manually control the motion of objects and record their movement using a pre-calibrated and fixed event camera (Prophesee EVK4). Subsequently, the tracking methods are validated using event streams and compared against other advanced algorithms. The object models are known a priori, and the ground truth of the pose is acquired using the OptiTrack system. The initial pose is obtained in a similar manner as described in [2].

In order to perform a quantitative evaluation of the errors associated with each method, we present the object pose tracking results using two well-established metrics [24]:

$$\delta_{\mathbf{R}} = \cos^{-1}\left(\frac{\text{trace}\left(\mathbf{R}^{\top}\mathbf{R}_{gt}\right) - 1}{2}\right), \quad (19)$$

$$\delta_{\mathbf{T}} = \left\|\mathbf{T} - \mathbf{T}_{gt}\right\|_2. \quad (20)$$

The evaluation of these two errors involves comparing the estimated rotation matrix $\mathbf{R}$ and translation vector $\mathbf{T}$ with their corresponding ground truth values $\mathbf{R}_{gt}$, $\mathbf{T}_{gt}$.

To evaluate the performance of the proposed method, we compare against state-of-the-art event-based methods. For objects with straight edges, we employ the following three baseline methods: *(i)* Line-Based. Inspired by the work of Chamorro et al. [2] on event camera line-SLAM, we employ the fundamental technologies presented in their method for object pose tracking. *(ii)* LS-based. The least squares-based approach optimizes the pose and tracks the object by minimizing the distance between events and lines. *(iii)* NNS-based. Initially, employ our method to extract corners and edges, then establish the correlation between corners and edges using the nearest neighbor search approach, for the purpose of comparison with our optical flow-guided method. For objects with curved edges, we compare our method with NNS-based methods.

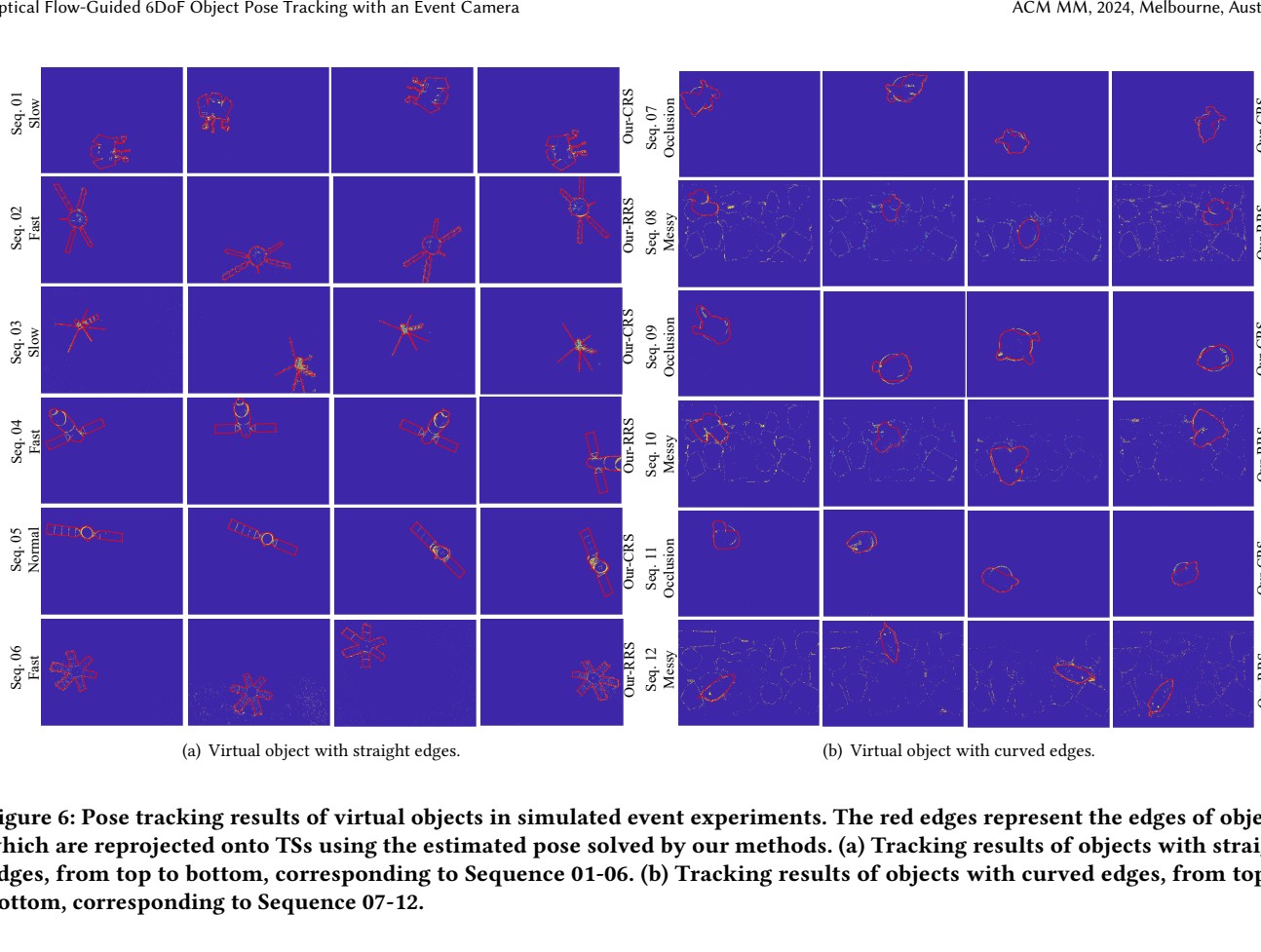

(a) Virtual object with straight edges.

(b) Virtual object with curved edges.

Figure 6: Pose tracking results of virtual objects in simulated event experiments. The red edges represent the edges of objects, which are reprojected onto TSs using the estimated pose solved by our methods. (a) Tracking results of objects with straight edges, from top to bottom, corresponding to Sequence 01-06. (b) Tracking results of objects with curved edges, from top to bottom, corresponding to Sequence 07-12.

Table 2: Pose tracking error of objects with curved edges in simulated event experiments ($\delta_{\mathbf{R}}$ :°, $\delta_{\mathbf{T}}$ : $cm$).

| Sequence | 07 | | | | | | 08 | | | | | | 09 | | | | | |
|---|---|---|---|---|---|---|---|---|---|---|---|---|---|---|---|---|---|---|
| | Original | | Messy | | Occlusion | | Original | | Messy | | Occlusion | | Original | | Messy | | Occlusion | |
| Method | $\delta_{\mathbf{R}}$ | $\delta_{\mathbf{T}}$ | $\delta_{\mathbf{R}}$ | $\delta_{\mathbf{T}}$ | $\delta_{\mathbf{R}}$ | $\delta_{\mathbf{T}}$ | $\delta_{\mathbf{R}}$ | $\delta_{\mathbf{T}}$ | $\delta_{\mathbf{R}}$ | $\delta_{\mathbf{T}}$ | $\delta_{\mathbf{R}}$ | $\delta_{\mathbf{T}}$ | $\delta_{\mathbf{R}}$ | $\delta_{\mathbf{T}}$ | $\delta_{\mathbf{R}}$ | $\delta_{\mathbf{T}}$ | $\delta_{\mathbf{R}}$ | $\delta_{\mathbf{T}}$ |
| NNS-Based | 4.12 | 6.44 | 2.89 | 8.92 | 3.54 | 8.24 | 3.51 | 7.47 | 3.87 | 10.11 | 4.87 | **8.25** | 2.71 | 7.24 | 4.21 | 9.68 | 3.91 | **8.26** |
| Ours | **2.08** | **5.21** | **2.62** | **7.51** | **2.56** | **7.62** | **2.12** | **7.21** | **2.65** | **8.68** | **2.22** | 8.87 | **2.14** | **6.87** | **3.57** | **7.98** | **2.86** | 8.47 |

| Sequence | 10 | | | | | | 11 | | | | | | 12 | | | | | |
|---|---|---|---|---|---|---|---|---|---|---|---|---|---|---|---|---|---|---|
| | Original | | Messy | | Occlusion | | Original | | Messy | | Occlusion | | Original | | Messy | | Occlusion | |
| Method | $\delta_{\mathbf{R}}$ | $\delta_{\mathbf{T}}$ | $\delta_{\mathbf{R}}$ | $\delta_{\mathbf{T}}$ | $\delta_{\mathbf{R}}$ | $\delta_{\mathbf{T}}$ | $\delta_{\mathbf{R}}$ | $\delta_{\mathbf{T}}$ | $\delta_{\mathbf{R}}$ | $\delta_{\mathbf{T}}$ | $\delta_{\mathbf{R}}$ | $\delta_{\mathbf{T}}$ | $\delta_{\mathbf{R}}$ | $\delta_{\mathbf{T}}$ | $\delta_{\mathbf{R}}$ | $\delta_{\mathbf{T}}$ | $\delta_{\mathbf{R}}$ | $\delta_{\mathbf{T}}$ |
| NNS-Based | 2.64 | 6.96 | 3.48 | 8.14 | 4.15 | 9.34 | 2.70 | 7.75 | 3.69 | 9.86 | 4.02 | 10.88 | 2.59 | 6.02 | 3.05 | 9.72 | **3.62** | 10.45 |
| Ours | **2.01** | **6.35** | **3.15** | **7.54** | **3.04** | **8.15** | **2.04** | **6.51** | **2.98** | **7.20** | **3.05** | **7.05** | **2.35** | **5.72** | **2.96** | **8.93** | 3.88 | **8.52** |

## 4.2 Simulated Event Experiments

Firstly, we conduct extensive experimental tests focusing on objects characterized by a notable abundance of lines. We utilize varying levels of object motion velocities, including slow, normal, and fast, to conduct a comparative analysis of methods. The pose tracking results are visually depicted in Figure 6(a), providing an intuitive display of the performance of the proposed methods. Based on the solved pose, the 3D point cloud of the object is projected onto the image plane while preserving its 2D edges. Moreover, we quantitatively evaluate the pose error of each method, as shown in Table 1. Overall, the proposed methods surpass the baselines in terms of accuracy. In sequence 03, there is a notable increase in errors across all methods. This can be attributed to the considerable distance between the object and the camera, the limited occurrence of events poses a challenge for pose tracking.

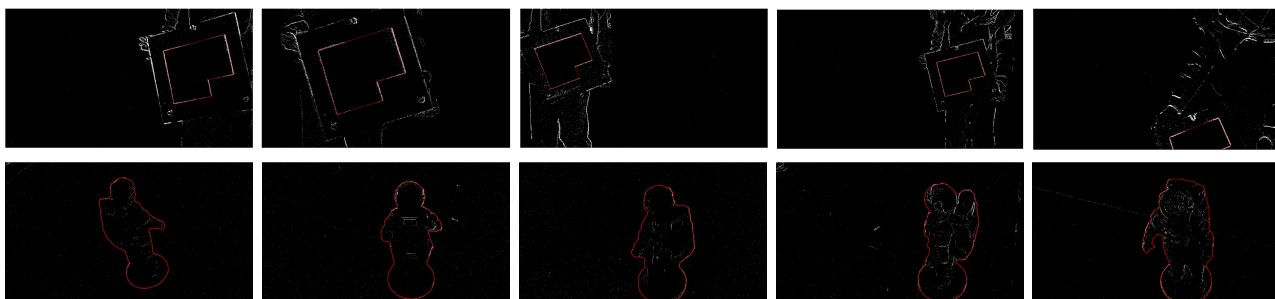

**Figure 7: Pose tracking results of the planar pattern and astronaut model in real event experiments. The red represents the edges of the 3D model, which are reprojected onto accumulated event images using estimated poses obtained from our methods.**

**Table 3: Pose tracking error of objects in real event experiments ($\delta_R : ^\circ$, $\delta_T : cm$).**

| Object | Planar pattern | | Cube | | Book | | Satellite model | | Astronaut model | |
|---|---|---|---|---|---|---|---|---|---|---|
| Type | Planar line | | Nonplanar line | | Planar line | | Nonplanar line | | Nonplanar curve | |
| Method | $\delta_R$ | $\delta_T$ | $\delta_R$ | $\delta_T$ | $\delta_R$ | $\delta_T$ | $\delta_R$ | $\delta_T$ | $\delta_R$ | $\delta_T$ |
| Line-Based | **1.16** | 2.99 | 1.44 | 3.81 | **1.35** | 3.94 | 3.02 | 6.58 | - | - |
| LS-Based | 2.46 | 3.49 | 1.46 | 4.14 | 1.39 | 3.91 | 2.98 | 5.94 | - | - |
| NNS-Based | 3.46 | 5.52 | 2.35 | 5.90 | 2.21 | 6.45 | 4.05 | 7.12 | 7.78 | 11.25 |
| Ours | 1.78 | **2.76** | **0.89** | **2.75** | 1.52 | **2.64** | **2.02** | **4.67** | **3.42** | **7.66** |

Next, we conduct tests on objects with curved edges. Since the edges of these objects are curved, the methods that rely on lines for pose tracking cannot be directly applied. However, our methods remain capable of consistently pose tracking of these objects, as depicted in Figure 6(b). To thoroughly validate the robustness of the methods, we also introduce challenging conditions such as messy backgrounds and object occlusions. Our methods both effectively address these challenges by establishing associations between corners and edges of objects. In Figure 6(b), the object edges and events exhibit a tight alignment utilizing the solved poses. We also quantitatively assess the pose tracking error of our methods, as demonstrated in Table 2. Our method yields smaller pose errors, as optical flow effectively guides the establishment of correlations between corners and curved edges. On the other hand, NNS-based methods solely seek out the nearest edge points, which can potentially lead to inaccuracies in matching corners with curved edges.

### 4.3 Real Event Experiments

Initially, we conduct real event experiments on objects with straight edges, including the planar pattern, cube, book, and satellite model. In comparative methods, partial lines of objects are employed for pose tracking. To ensure a fair comparison, our methods utilize corresponding partial lines as object edges for tracking. The definition of pose error is consistent with that in the previous test using Eqs. (19) and (20), and the results are shown in Table 3. The tracking outcome of the planar pattern is illustrated in Figure 7. The color red denotes the edges of the planar pattern, which are projected onto accumulated event images utilizing the solved poses obtained

through our methods. Experimental results demonstrate that our methods outperform the compared methods in terms of accuracy.

We select the astronaut model as a representative subject for conducting object pose tracking experiments. In contrast to the previous experiment, this time the astronaut model remains stationary while the event camera is moved. This experimental setup enables us to test both the scenario of a stationary model with a moving camera and the scenario of a stationary camera with a moving model. We conduct statistical analysis on the pose tracking errors, as presented in Table 3. It is observed that our methods achieve higher accuracy. The pose tracking results are visually displayed in Figure 7. The projected edges of the astronaut model remain tightly aligned with events throughout the entire tracking process, providing an intuitive demonstration of the accuracy of our tracking methods.

## 5 CONCLUSION

This paper proposes an optical flow-guided 6DoF object pose tracking method using an event camera. Firstly, a hybrid feature extraction strategy is employed to detect corners from events and extract edges from the projected point cloud. Then, the optical flow of corners is calculated based on the spatio-temporal probability distribution of events. Subsequently, we establish associations between corners and edges along the direction of the optical flow. Furthermore, by minimizing the distance between edges and corners, an iterative pose refinement procedure is employed for continuous tracking of objects. The experimental results of both simulated and real events indicate that our methods outperform the state-of-the-art event-based methods, particularly in challenging scenarios involving severe occlusion and messy backgrounds.

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
