# OpenReview forum: "Optical Flow-Guided 6DoF Object Pose Tracking with an Event Camera"
_acmmm.org/ACMMM/2024/Conference — MM2024 Poster_

### Official Review · Reviewer_GcKA · 2024-05-24

**Rating:** 3
**Confidence:** 4

**Summary:**

This paper introduces a method for tracking the 6DoF pose of an object using an event camera guided by optical flow.

**Strengths:**

1. The paper is easy to understand, and the pictures are well depicted.
2. The paper method has a large number of formulas described and is relatively interpretable.

**Limitations:**

1. It is well known that traditional feature-matching methods are very time-consuming, and the paper lacks a description of the hardware performance of this algorithm, e.g., inference speed, memory footprint, etc.

2. Algorithm 1 does not look like pseudo-code, so please write carefully.

3. The results of the comparison with learning based are missing from the experiment, and this approach is described in realted work.

4. It is recommended to add event-based 6 DOF pose estimation with a learning-based method to the relevant literature.[1][2]

5. The authors in the rebuttal are going to focus on why using traditional methods and not applying learning-based methods, such as learning-based method MSE error  is poor or something like that.

6.  The authors mention noise effects in line 109, but the datasets demonstrated are ideal, and the actual environments have very strong hot and background noise, and I'm curious if the algorithm will be able to adapt to these environments very robustly.

[1] Nguyen A, Do T T, Caldwell D G, et al. Real-time 6DOF pose relocalization for event cameras with stacked spatial LSTM networks[C]//Proceedings of the IEEE/CVF Conference on Computer Vision and Pattern Recognition Workshops. 2019: 0-0.
[2] Lin H, Li M, Xia Q, et al. 6-DoF Pose Relocalization for Event Cameras With Entropy Frame and Attention Networks[C]//Proceedings of the 18th ACM SIGGRAPH International Conference on Virtual-Reality Continuum and its Applications in Industry. 2022: 1-8.

**Suitability:**

1

---

### Official Review · Reviewer_wTHU · 2024-05-24

**Rating:** 5
**Confidence:** 4

**Summary:**

This paper proposes a novel method to track the 6DoF object pose from the event camera. The event camera is more robust in motion blur than the traditional camera. This paper proposes a flow-guided and a two-stage pipeline. They first extract 2D-3D hybrid features and establish the correlation between corners and edges guided by optical flow. The final 6DoF pose is calculated by minimizing the distances between corners and edges.

**Strengths:**

1. The motivation is reasonable, and the method and experiment are solid, the writing is good.
2. The proposed method is useful in real-world downstream applications, such as self-driving and robotics.
3. The paper also conducts a thorough comparison using simulated and real events.

**Limitations:**

1. The authors did not provide information about the running speed.  It's important to evaluate if your method can be used in real-time tracking.
2. The failure cases are missing. I don't know the limitations of your method.
3. The experiment is not enough. I suggest to compare with one or two SOTA methods of 6D tracking using a traditional camera. These additional experiments can thus demonstrate that the event camera is really useful. Such as, 6-PACK.
[1] 6-PACK: Category-level 6D Pose Tracker with Anchor-Based Keypoints

**Suitability:**

3

---

### Official Review · Reviewer_DEWR · 2024-05-25

**Rating:** 4
**Confidence:** 3

**Summary:**

This paper proposes an optical flow-guided 6DoF object pose tracking method using an event camera. Specifically, the authors begin with a hybrid feature extraction strategy that detects corners from events and extracts edges from the projected point cloud. The optical flow of these corners is then calculated based on the spatiotemporal probability distribution of events. By associating corners and edges along the direction of the optical flow, the method establishes robust connections. The object pose is continuously tracked through an iterative refinement procedure that minimizes the distance between edges and corners. Experimental results, both simulated and real-world dataset, demonstrate that this method outperforms existing state-of-the-art event-based techniques, especially in challenging scenarios with severe occlusion and cluttered backgrounds.

**Strengths:**

1. The use of an event camera to address the problem of 6DoF object pose tracking in extreme conditions (e.g., high speed or low light) is both practical and innovative.
2. The paper is well-written, clear, and easy to read.
3. The authors made a significant effort in creating simulated and real 6DoF object pose tracking datasets, which is highly valuable for researching this novel problem.

**Limitations:**

1. The comparison methods presented are somewhat limited. Although there are few works on 6DoF object pose tracking using event cameras, and even fewer open-source projects related to event cameras, there are many traditional RGB deep learning approaches. The authors are encouraged to test some classical deep learning open-source codes on both simulated and real datasets.
2. The descriptions of the simulated and real datasets are insufficient in the main manuscript. This is a significant contribution by the authors to the field. Although some details about the datasets are provided in the appendix, the authors should summarize the contributions and describe the datasets more thoroughly in the main manuscript.
3. As this work focuses on optical flow-guided 6DoF object tracking, the authors should include more visualizations related to optical flow in the main text, especially comparative visual results with and without optical flow.
4. The authors should demonstrate the tracking accuracy over time, including both quantitative and visual experiments.
5. The major advantage of event cameras is their ability to handle high-speed motion blur and low-light scenarios for 6DoF pose tracking. The authors should provide examples of extreme scenarios, particularly comparing them with RGB scenarios. Although the authors used the single-modality EKV4 camera, future work could explore multimodal cameras, using beam splitters to simultaneously capture RGB and event streams, or using multi-camera setups with spatiotemporal calibration.
6. The writing could be further improved, including the details in the text and figures. For example, the backgrounds in Fig. 6 and Fig. 7 should be consistently presented.

**Suitability:**

2

---

### Meta-Review · Area_Chair_eZr9 · 2024-06-26

**Recommendation:** Accept (Poster)
**Confidence:** 3

**Metareview:**

This paper proposes an innovative optical flow-guided 6DoF object pose tracking method using an event camera. The method demonstrates robustness in challenging conditions, such as high-speed motion and low light, making it a practical contribution to the field. The authors provide extensive experimental validation on both simulated and real-world datasets, showing superior performance over existing event-based techniques.

However, the paper would benefit from more comparisons with traditional RGB deep learning approaches and further descriptions of the datasets used. Additionally, the impact of event camera advantages, such as handling high-speed motion blur and low-light scenarios, could be better highlighted with more visual examples.